



# Detecting ground ice in warm permafrost with the dielectric relaxation time from SIP observations

Hosein Fereydooni[1,*], Stephan Gruber[2,*], David Stillman[3,*], and Derek Cronmiller[4,*]

[1]Department of Earth Sciences - Carleton University, Ottawa, Ontario, Canada
[2]Department of Geography and Environmental Studies- Carleton University, Ottawa, Ontario, Canada
[3]Southwest Research Institute, Boulder, CO, USA
[4]Permafrost Geologist, Energy, Mines and Resources, Yukon Geological Survey, Yukon, Canada
[*]These authors contributed equally to this work.

**Correspondence:** Stephan Gruber (stephan.gruber@carleton.ca)

**Abstract.** The melting of ground ice poses significant hazards in permafrost regions, making effective detection methods essential. Conventional geophysical techniques like electrical resistivity, seismic surveys, or ground-penetrating radar alone often produce ambiguous results due to the overlap of material characteristics between frozen and unfrozen ground. This study addresses these limitations by using the dielectric relaxation time of ice as a unique indicator of ground ice. We developed a
method to quantify relaxation time from Spectral Induced Polarization (SIP) data measured by the FUCHS III device. The method's effectiveness was demonstrated through synthetic data and two field surveys. SIP field measurements, ranging from 1.46 Hz to 40 kHz, were conducted on a retrogressive thaw slump and a pingo in Yukon, Canada. The extracted relaxation times were mapped to pseudo-depths obtained from single-frequency inversion. This study proposes a relaxation time range from 10 to 400 $\mu$s for ground ice, and the results demonstrate that this range can detect ground ice spectra in field studies.
Comparison with observations in a borehole and an exposure of permafrost indicate that relaxation time is less ambiguous in detecting ground ice in warm permafrost than conventional methods such as electrical resistivity tomography.

Keywords: Ground ice detection, dielectric relaxation time, spectral induced polarization, permafrost

## 1 Introduction

The ability to detect and characterize ground ice in permafrost is important because ground-ice melt results in hazards that affect infrastructure, ecosystems, hydrology, as well as traditional livelihoods and cultural practices. Without knowledge of the distribution and amount of ground ice, corresponding hazards arising from disturbance and climate change cannot be anticipated.

Where direct observation with drilling or excavation is impractical or too expensive, established geophysical methods such as
Electrical Resistivity Tomography (ERT, e.g., Olhoeft, 1977; You et al., 2013; Oldenborger, 2021; Li et al., 2021; Farzamian et al., 2024), Ground Penetrating Radar (GPR, e.g., Brandt et al., 2007; You et al., 2013), and seismic methods (e.g., Musil et al., 2002; Tourei et al., 2024) are commonly used to infer the presence of ground ice. Interpreting these data, however,



can involve ambiguity, as the characteristics of frozen ground and other materials partially overlap. Joint interpretation and inversion (Hauck et al., 2011; Wagner et al., 2019) can overcome these limitations in some areas, albeit at additional cost and
effort.

By contrast, the dielectric relaxation time of ice has the potential for affording confident single-method detection of ground ice. To this end, several studies have employed Spectral Induced Polarization (SIP) in recent years (Huisman et al., 2016; Maierhofer et al., 2022; Mudler et al., 2022). SIP instruments for measuring impedance spectra in the field exist, but there is little experience with survey design and processing. Since methods for inverting complete SIP observations are not readily
available, the interpretation of SIP measurements has often been limited to using few individual frequencies (Grimm and Stillman, 2015; Fereydooni et al., 2024).

Here, we develop and demonstrate an alternative approach: We interpret the measured spectra directly to interpret the presence or absence of the diagnostic ice relaxation with confidence. For this, we forgo the level of geometric detail of a tomography, which we only use to establish the approximate location points at pseudo depths based on one frequency.

In order to improve the geophysical identification of ground ice, the goal of this paper is to reduce the ambiguity in the detection of ground ice via SIP, even in the absence of tools for full tomographic inversions. Our objectives are: (1) To establish ranges of relaxation times that can be attributed confidently to ground ice. (2) To develop a method for quantifying relaxation time from SIP field data. (3) To outline the conditions under which such relaxations can be used to infer ice at the approximate pseudo-location of a single observation. And (4), to demonstrate the overall approach based on field observations in warm and
ice-rich permafrost.

Relevant relaxation frequencies will be identified by reviewing the dielectric behaviour of ice and frozen ground as well as observations from a range of frozen and unfrozen materials. The method for obtaining relaxation time will use the Debye model and optimization techniques, and the appropriate conditions will be characterized based on geophysical theory. The demonstration cases use late-winter measurements from a retrogressive thaw slump and a pingo in Yukon, Canada.

## 2   Background: dielectric sensing and material characteristics at $1$–$10^5$ Hz

### 2.1   Ice

In the presence of an alternating current (AC) field, protonic point (Bjerrum) and ionic defects in ice facilitate effective charge transport within the crystal lattice. These defects allow protons to reorient water molecules, resulting in ice polarization and affecting its dielectric properties (Petrenko and Whitworth 1999). This polarization affects electrical impedance, a fundamental
aspect of electrical circuits, which encompasses both resistance and capacitance and which manifests phase differences between input voltage and output current. In principle, electrical impedance reflects a circuit's ability to resist current flow (real part, resistivity) and to store electrical energy (imaginary part, reactance). SIP is a geophysical technique that involves measuring the electrical impedance of subsurface materials at multiple frequencies $Z^*(\omega)$:



$$Z^*(\omega) = \frac{V(t)}{I(t)} = Z_A e^{i\phi} = Z_A(\sin(\phi) + i\cos(\phi)) = Z_{\text{Real}}(\omega) + iZ_{\text{Imag}}(\omega) \tag{1}$$

$$\phi = \arctan\left(\frac{Z_{\text{imag}}}{Z_{\text{real}}}\right) \tag{2}$$

Where $t$ is time, $i = \sqrt{-1}$, $V(t)$ is the potential in volts, $I(t)$ is the current in amperes, $\omega$ is the angular frequency ($\omega = 2\pi f$, and $f$ is frequency in Hz), $Z^*(\omega)$ ($\Omega m$) is the electrical impedance spectrum, $Z_{\text{real}}(\omega)$ and $Z_{\text{imag}}(\omega)$ are the real and imaginary parts of the impedance spectrum, respectively, and $\phi$ is the phase shift defined by Eq.(2).

When a material is subjected to an AC field, the relaxation time refers to the time required for dipolar molecules to reorient
themselves in response to the changing electric field. This concept was introduced by Maxwell and expanded by Debye (Eq.(3)), who quantified it as the relaxation time ($\tau$) inversely related to the critical relaxation frequency (Cole and Cole, 1941).

$$\varepsilon'(\omega) = \varepsilon_\infty + \frac{\varepsilon_s - \varepsilon_\infty}{1 - i\omega\tau} \tag{3}$$

Where $\varepsilon'(\omega)$ is the real part of dielectric permittivity, $\varepsilon_\infty$ is the dielectric permittivity at the highest frequency, and $\varepsilon_s$ is the dielectric permittivity at the lowest frequency. The Debye equation can be rewritten for the real and imaginary parts as follows:

$$\varepsilon'(\omega) = \varepsilon_\infty + \frac{\varepsilon_s - \varepsilon_\infty}{1 + (\omega\tau)^2} \tag{4}$$

$$\varepsilon''(\omega) = \varepsilon_\infty + \frac{\omega\tau(\varepsilon_s - \varepsilon_\infty)}{1 + (\omega\tau)^2} \tag{5}$$

The Debye model can be reformulated using electrical impedance:

$$Z^*(\omega) = \rho_0\left\{1 - m\left[1 - \frac{1}{1 + (i\omega\tau)^c}\right]\right\} \tag{6}$$

where $\rho_0$ is the real apparent resistivity (see Preprocessing section) at low frequencies, $m$ is the chargeability, and $\tau$ is the
relaxation time (with the relaxation frequency given by $\frac{1}{2\pi\tau}$).

## 2.2 Relaxation Time

Relaxation time is a unique fingerprint for detecting ice. The dependency of the relaxation time on the temperature can be explained using the Arrhenius equation (Artemov and Volkov, 2014), as shown in Eq.(7):

$$\tau(T) = \alpha\exp\left(\frac{E_0}{kT}\right) \tag{7}$$



**Table 1.** Perfect pure ice (with $E_0 = 0.58$ eV) relaxation time at different temperatures.

| Temperature (°C) | 0 | -23 | -40 |
|---|---|---|---|
| Relaxation Time ($\tau$ in $\mu$s) | 22 | 450 | 3,600 |

Where $T$ is temperature in Kelvin, $\alpha$ is the pre-exponential factor (a constant specific to the material), $E_0$ is the activation energy, and $k$ is the Boltzmann constant ($8.617333 \times 10^{-5}$ eV/K).

The Arrhenius equation expresses that $\tau$ is dependent on temperature, which is due to the molecular dynamics and proton diffusion in ice. As temperature decreases, the relaxation time of ice increases exponentially, driven by the reduced mobility of Bjerrum defects and the higher energy barriers for molecular reorientation (Petrenko and Whitworth, 1999; Stillman et al.,
2013a, b).

At 0°C, ice has a relaxation time of 22 $\mu$s, while water shows $1.7 \times 10^{-5}$ $\mu$s. As the temperature decreases to -23°C, the relaxation time increases to approximately 450 $\mu$s (Table 1). This trend continues, with relaxation times growing exponentially as the temperature drops further (Artemov and Volkov, 2014; Mudler et al., 2022).

The dielectric behavior of ice at 0°C is primarily influenced by the concentration and mobility of Bjerrum L-defects. While
the dissociation of $H_2O$ molecules does produce ionic defects ($H_3O^+$ and $OH^-$), and these contribute significantly to the DC electrical conductivity of ice, the primary Debye dielectric relaxation mechanism in ice is attributed to the reorientation of water molecules facilitated by the movement of Bjerrum defects, particularly L-defects (Stillman et al., 2013a, b).

The activation energy associated with the mobility of these L-defects governs the temperature dependence of the relaxation time. While perfectly pure ice theoretically has a high activation energy for intrinsic L-defects (around 0.58 eV), real ice
always contains impurities that create extrinsic defects, leading to lower effective activation energies. The activation energy for L-defects can range from approximately 0.19-0.24 eV at low to moderate concentrations or low temperatures, increasing to around 0.294 eV at higher concentrations or very low temperatures, such as in $Cl^-$ saturated ice (Stillman et al., 2013a). The diffusion coefficient of protonic defects (including L-defects) in ice is significantly smaller than that of ions in liquid water, which affects the kinetics of the relaxation process and leads to a longer relaxation time ($\tau$) in ice compared to water at 0°C.
However, the key factor determining the primary dielectric relaxation in ice is the mobility (and hence diffusion) of Bjerrum L-defects, which allows for the reorientation of water dipoles (Petrenko and Whitworth, 1999).

## 2.3   Mixed media

The dielectric relaxation of ice exhibits distinct frequency ranges, activation energies, and compositional dependencies that allow it to be easily distinguished. Its high permittivity, long relaxation time, and sensitivity to impurities and silica surfaces
make dielectric relaxation a powerful way for detecting and analyzing ice.

Stillman et al. (2010) identified five distinct dielectric relaxations in ice-silicate mixtures, each with unique properties that may facilitate their differentiation from ice relaxation. The fastest is the orientational polarization of adsorbed water, occurring at



630 kHz at -92.15°C with an activation energy of 0.4 eV. This is followed by the orientational polarization of ice, caused by L-Bjerrum defect rotation, with a 0.29 eV activation energy. The adsorbed water Maxwell-Wagner interfacial polarization occurs

at frequencies below ice relaxation at -90.15°C, also with a 0.4 eV activation energy. At much lower frequencies, around 10 mHz at -92.15°C, the hydrate-silicate Maxwell-Wagner interfacial polarization takes place. Finally, the Low-Frequency Dispersion (LFD) is observed in samples with less than 2vol% $H_2O$, attributed to charge hopping within the adsorbed water layer.

Also, the relaxation time of mixed frozen soils (Appendix Table A1, Bittelli et al. (2004)) differs from pure ice (Table 1) due

to several factors. Impurities, particularly chloride ions, create additional defects in the ice structure, leading to faster relaxation times. Silica surfaces in soil mixtures influence the defect density in adjacent ice, possibly due to stronger electric fields or chloride supersaturation near pore walls. As ice content decreases, the relaxation time decreases, suggesting a higher relative density of lattice defects in smaller ice volumes (Stillman et al., 2010). These factors combine to create a complex dielectric response in frozen soil mixtures, often resulting in multiple relaxation processes and broader relaxation distributions compared

to pure ice, which is crucial for accurately interpreting geophysical measurements and characterizing subsurface ice content.

The relaxation time of graphite-rich soils (Revil et al., 2019) can overlap with the typical relaxation time range of ground ice, which may lead to ambiguous interpretations. Therefore, when conducting geophysical surveys in areas containing graphitic materials, this potential overlap must be carefully considered during data analysis and interpretation to avoid misidentification of ground ice.

## 2.4  Interpretation of impedance measured in heterogeneous ground

We can use a forward model to visualize the relaxation times of frozen soil samples, and unfrozen materials. By keeping the other parameters ($\rho_0$, $m$, $c$) constant and varying only the relaxation time ($\tau$) in Eq.(6), it becomes clear how the impedance signal can be used to differentiate ground ice from other materials (Figure 1).

To accurately identify the ground-ice signal, certain criteria must be followed to ensure that the SIP spectrum can be recognized

as a ground ice signature. First: according to the equivalent ice circuit (Debye circuit), the real part of ground ice (SIP) signal decreases with increasing frequency ($Z_{\text{Real}}(\omega) = \frac{\omega^2 R_1 C_1^2}{(\omega^2 R_1 C_1 C_2)^2 + \omega^2 (C_1 + C_2)^2}$). Second: The imaginary part should exhibit a peak at relaxation time of ice ($Z_{\text{Imag}}(\omega) = \frac{\omega^3 R_1^2 C_1^2 C_2 + \omega (C_1 + C_2)}{(\omega^2 R_1 C_1 C_2)^2 + \omega^2 (C_1 + C_2)^2}$). Third: The phase angle should display negative values, reflecting the role of ice as a capacitance. Fourth: The obtained relaxation time should fall within the expected range for ground ice.

# 3  Methods and materials

## 3.1  Preprocessing

The FUCHS III+ device measures impedance magnitude ($\Omega$) and phase angle (in degree). So, as first step, we need to convert the measured impedance to apparent impedance by using the geometric factor ($K_g$) ($Z^*_{apparent} = Z^*_{measured} \times K_g$).





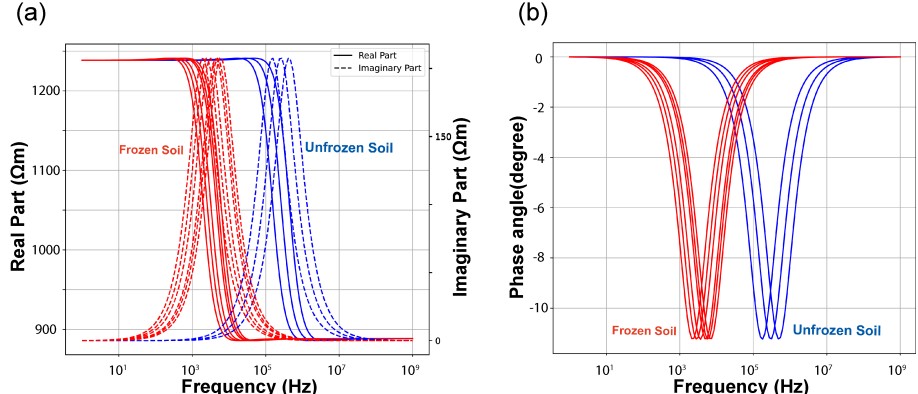

**Figure 1.** Synthetic frozen (red, temperatures: -20°C, -15°C, -10°C, -5°C, -1°C), and unfrozen (blue, temperature: 38°C, 46°C, 57°C) soil SIP spectra: (a) Real and imaginary part, (b) phase angle.

$$K_g = \frac{2\pi}{\left[\frac{1}{\overline{AM}} - \frac{1}{\overline{AN}} - \frac{1}{\overline{BM}} + \frac{1}{\overline{BN}}\right]} \tag{8}$$

Where: A and B are current electrodes; M and N are potential electrodes, $\overline{AM}$, $\overline{AN}$, $\overline{BM}$, and $\overline{BN}$ represent the distances between respective electrodes.

The measured spectra need to be assessed for spike noise, characterized here by a standard deviation exceeding +/- 3 away from the mean. These may occur at one or more frequencies. In such cases, the code allows excluding these measurements, which is crucial because it reduces the sensitivity to the initial guess and allows for a broader range of values for each unknown

parameter, rather than a narrow one (Appendix Fig. A1). If a positive phase angle is detected, it will be investigated whether these values result from unfrozen material or are indicative of spike noise, and it will follow the same procedure outlined for handling spike noise.

### 3.2    Extracting relaxation times

Extracting relaxation times is one of the avenues for processing SIP data to allow interpretation. It is challenging, because

measured spectra have noise that can complicate the extraction of parameters, and more than one relaxation feature may affect a frequency band of interest. In this study, we fit a multi-Cole-Cole model (Eq. (9)) to SIP spectra to extract seven unknown parameters for the two most prominent relaxation frequencies (Pelton et al., 1978).

$$\rho(\omega) = \rho_0 \left\{ 1 - m_1 \left[ 1 - \frac{1}{1 + (i\omega\tau_1)^{c_1}} \right] \right\} \left\{ 1 - m_2 \left[ 1 - \frac{1}{1 + (i\omega\tau_2)^{c_2}} \right] \right\} \tag{9}$$

Where $\omega$ represents the angular frequency, $\rho(\omega)$ is the complex apparent resistivity, $\rho_0$ is apparent resistivity at low fre-

quency, $m_1$ and $m_2$ are the chargeabilities, $\tau_1$ and $\tau_2$ are the relaxation time, $c_1$ and $c_2$ are the frequency coefficients.





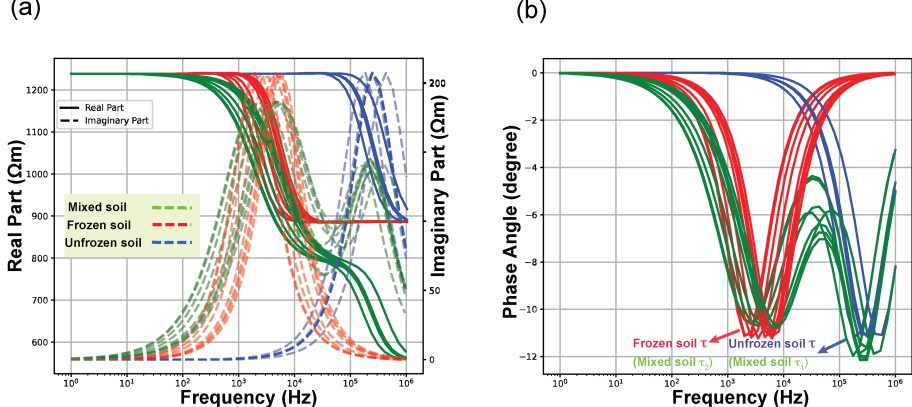

**Figure 2.** Synthetic data showing (a) the real and imaginary parts and (b) the phase angle of unfrozen (blue), frozen (red), and mixed (frozen and unfrozen, green) soil.

Utilizing the multi-Cole-Cole model, which extracts two relaxation times here, allows for identifying relaxation times of mixed materials, such as soils. For instance, the combination of unfrozen soil samples (blue line in Fig. 2) and frozen soil samples (red line in Fig. 2, Appendix Table A2) that cannot be explained by one single relaxation time (green line in Fig. 2 and Eq.(6)).

We utilized a general least square method to minimize the difference between measured SIP spectra, and data predicted by the multi-Cole-Cole model (Eq.(9)) to find seven unknown parameters ($\rho_0$, $m_1$, $m_2$, $\tau_1$, $\tau_2$, $c_1$, and $c_2$). For this process, we developed a Python script. It first converts the field-measured impedance to apparent impedance by using geometric factor, then assigns a weight to the real and imaginary part of data to ensure those components are fairly balanced and manages noisy data using device-reported error percentages.

The two extracted relaxation times ($\tau_1$, $\tau_2$) will then be compared to the ground ice relaxation time range. If either of them falls within this range, it will be identified as ground ice SIP spectrum.

In this paper, the accuracy of the results is presented using the Root Mean Square Error (RMSE) percentage:

$$\text{RMSE Percentage} = \frac{\sqrt{\frac{1}{n}\sum_{i=1}^{n}(|Z|\text{measured},i - |Z|\text{predicted},i)^2}}{\overline{|Z|}} \times 100 \tag{10}$$

where $|Z|_{\text{measured},i}$ and $|Z|_{\text{predicted},i}$ are the measured and predicted impedance magnitudes for the $i$-th frequency, respectively, and $\overline{|Z|}$ is the mean of the measured impedance magnitudes.

While the RMSE percentage can provide useful insights for determining the reliability of the results of the spectrum fit from the code, the results still need to be evaluated by an expert because the optimization process might get trapped in a local minimum, producing a low RMSE but not necessarily the best possible fit. An expert can recognize when this occurs by examining the overall shape of the fitted curves and their physical plausibility.



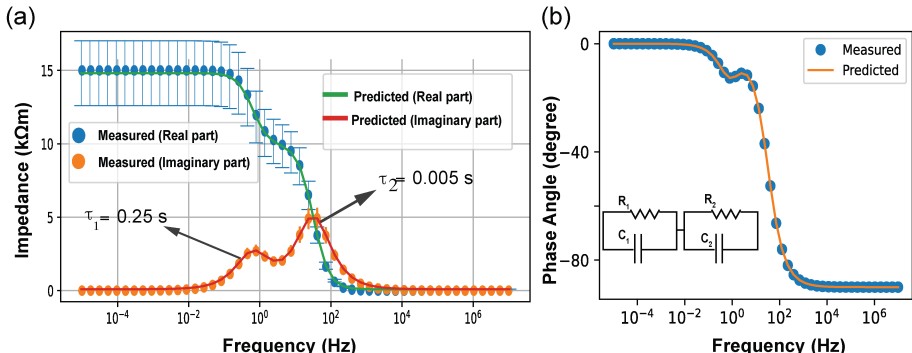

**Figure 3.** Comparison of Voigt model and predicted results. (a) Real and imaginary components, and (b) Phase angle. Dots represent exact Voigt model values and lines show predicted values.

The apparent relaxation time (ART) is then assigned a pseudo depth calculated with the ResIpy software package (Blanchy et al., 2020) to provide a 2D image of the distribution of ART versus pseudo-depth. The concept of pseudo-depth in electrical resistivity surveys represents an approximation of the depth range from which the majority of the measured signal originates. Typically, about 70-80% of the signal comes from this range, although this can vary depending on the array geometry and subsurface conditions (Edwards, 1977). The vertical sensitivity of these measurements follows a bell-curve distribution, with the peak contribution occurring at the pseudo-depth (Roy, 1974).

### 3.3   Code testing

We evaluate the method and code using synthetic data from a Voigt model, which represents multiple relaxations consisting of parallel resistor-capacitor (RC) elements, each representing a distinct relaxation mechanism. We use $R_1 = 1\,\mathrm{k\Omega}$, $R_2 = 10\,\mathrm{k\Omega}$, $C_1 = 50\,\mathrm{mF}$, $C_2 = 0.5\,\mathrm{mF}$. To simulate imperfect observations, we add five percent uniform noise.

The relaxation times predicted from the proposed code are $\tau_1 = 0.25$s and $\tau_2 = 0.005$s with RMSE=$3.5 \times 10^{-5}$%, which match the exact relaxation times of the Voigt model $\tau_1 = R_1 \times C_1 = 0.25$s, and $\tau_2 = R_2 \times C_2 = 0.005$s as shown in Fig. 3. This demonstrates the ability of the proposed code to extract the relaxation times.

### 3.4   Instrumentation

A FUCHS III+ device was employed for SIP measurements. It operates in the frequency domain and consists of a base unit to control measurements, one current unit that injects current into the ground, and three potential units that measure the potential change, with one potential unit acting as a reference to minimize local noise interference. The base unit and the other remote units are connected by fiber-optic cable. The device utilizes sinusoidal current (400 V, max. 1.5 mA) for impedance measurement.




Field spectra were acquired at 20 frequencies from 1.46 Hz to 40 kHz using a dipole-dipole configuration and locally appropriate spacing and expansion factors. The ground electrodes are located centrally between the active potential and current electrode pairs. A geofabric ground cover is used to ensure the smooth movement of fiber optic cables.

The real part of impedance measured at the lowest frequency (1.46 Hz) is considered equivalent to ERT observations because at this frequency, the system behaves similarly to a DC circuit. In this near-DC state, the imaginary component of impedance becomes negligible, and the measured impedance primarily reflects the real part - the resistivity. This closely mimics ERT measurements, which use DC or very low-frequency currents to measure subsurface resistivity. By focusing on the real part at such a low frequency, the measurement avoids the complexities introduced by phase angle changes and reactive components, providing a direct parallel to standard ERT results.

## 4 Field observations

### 4.1 Retrogressive thaw slump

Retrogressive thaw slumps (RTS) result from the thawing of ice-rich permafrost, featuring a steep headwall that gradually recedes due to surface thawing while periodic slurry flows carry away thawed sediment (Burn and Lewkowicz, 1990). They are typically found in massive ice bodies or ice-rich silts along lakes, rivers, and coasts and can pose risks to infrastructure. We performed SIP measurements on a RTS above the Takhini River at km 1,456.45 of the Alaska Highway (Figure 4). It initialized in 2014 and retrogressed towards the highway, with cracks close to the road embankment observed in 2019. Research in 2021 revealed further retrogression of the headwall and substantial sediment loss into the river. The headwall (Figure 4(c)) revealed a 1.5-metres-thick layer of eolian sand, over a 0.5–1.0 m thick layer of stratified fluvial sand, and an underlying layer of ice-rich glaciolacustrine silt and clay (Roy et al., 2021). In 2024, the highway was locally rerouted.

The surrounding area is characterized by mixed forest dominated by white spruce and aspen, with shrubs like willow and soapberry, along with other plant species such as fireweed and alpine sweet vetch. The site's history includes a 1958 wildfire, likely responsible for the thin organic cover. Permafrost in the region is sporadic and discontinuous, often ice-rich and sensitive to thawing. Ground temperatures just below 0°C and exposed ice-rich permafrost suggest a colder and wetter past environment. Cryostratigraphic analysis indicates syngenetic permafrost with gentle thermal gradients and ample water supply during its formation (Roy et al., 2021).

A long SIP survey was conducted on the surface above the RTS headwall (Figure 4, $1^{st}$ electrode: 135.51552 W, 60.85612 N, and $49^{th}$ electrode: 135.51467 W, 60.85629 N) and approximately parallel to the highway on 17–19 March 2023. During this period, local air temperatures ranged from -3°C in the morning to +6°C in the evening. The measurements followed a 50-metre profile, with one meter spacing between electrodes, and a spacing factor (a) from 2 to 5 and an expansion factor (n) from 1 to 8. In total, 464 impedance spectra were recorded.

Approximately 20 meters from the main survey line, two additional 2-metre (short) surveys were conducted directly in the exposed ice-rich material of the RTS headwall (Figure 4(c)). The perpendicular surveys were aligned horizontally, along the



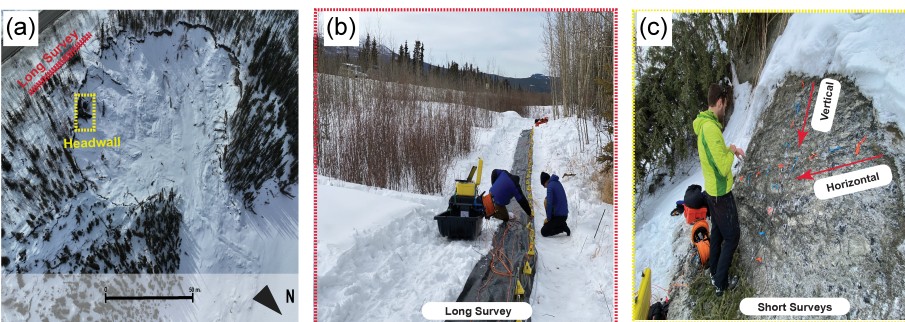

**Figure 4.** (a) The retrogressive thaw slump, headwall, (b) long and (c) short SIP survey.

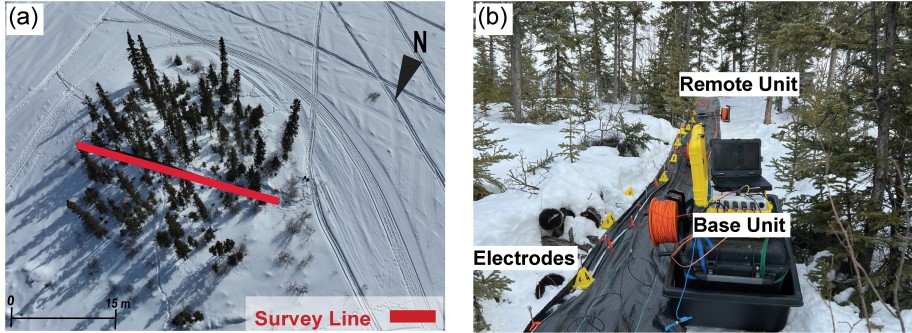

**Figure 5.** (a) The pingo site and (b) the survey line with the FUCHS device.

220  near-horizontal ground ice layers, and vertically. With 25 cm spacing between electrodes, and a spacing factor (a) from 2 to 5 and an expansion factor (n) from 1 to 8. In total, 37 impedance spectra were recorded in the horizontal and vertical profiles.

### 4.2 Pingo

Pingos are raised, dome-like geomorphic features found in areas with permafrost. They form when pressurized ground water creates a large ice body beneath the surface, causing the frozen ground above to rise. Pingos can stand over 50 meters high,

225  with a base that is typically circular (Flemal, 1976). An SIP survey was conducted on a pingo with a height of 3.5 meters (Figure 5, $1^{st}$ electrode: 137.53228 W,60.78356 N, and $29^{th}$ electrode: 137.53196 W, 60.78372 N) in Haines Junction, Yukon, Canada on 20–22 March 2023. During the measurements, local air temperatures ranged from -2°C in the morning to +4°C in the evening. The 30-meter survey line was measured with spacing factors (a) from 1 to 5 and an expansion factor (n) from 1 to 8. In total, 582 SIP spectra were collected (Figure 5(b)).



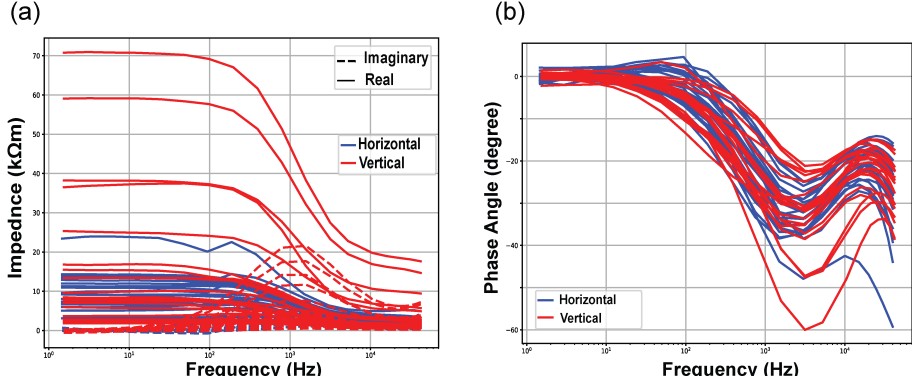

**Figure 6.** Horizontal (blue) and vertical (red) SIP spectra surveys in the headwall showing: (a) real (solid line) and imaginary (dashed line) parts, and (b) phase angle.

## 5    Results and interpretation

### 5.1    Retrogressive thaw slump: short profiles in headwall

The surveys made directly in the ice-rich permafrost allow testing our method without the additional effect of an unfrozen layer between the electrodes and the frozen ground of interest. In the spectra obtained (Figure 6), the real part dominates at frequencies below 100 Hz, consistent with the general trend of a stronger real part at lower frequencies. At intermediate frequencies (100 Hz to 10 kHz), the proportion of the real part decreases while the proportion of the imaginary part increases. The spectra for the vertical survey exhibit higher impedance (real and imaginary) than the horizontal one (Figure 6(a)). This is because in the vertical survey the current flow is perpendicular, and in the horizontal one it is parallel to the near-horizontal layering of segregated ice and soil. The phase angles are approximately the same for both surveys.

Relaxation times have been calculated for all spectra. The average RMSE from the fitting procedure was 4.6% for the horizontal and 2.0% for the vertical survey. From the two extracted relaxation times, those falling within the typical range of ground ice (Tables 1, A1, and A2) were selected as the ground ice relaxation time (72–253$\mu$s for the horizontal and 69–392$\mu$s for the vertical survey) (Figure A2 showing one spectrum fitting).

### 5.2    Retrogressive thaw slump: long profiles above headwall

We first consider the individual frequency inversion results obtained using the ResIpy package. The result at 1.46 Hz (like ERT) indicates a resistive region in the center of the survey area (Figure 7(a)). The inversion result for the real part at 40 kHz (Figure 7(b) shows higher resistivity values in the western and eastern parts of the profile, although most resistivity values remain below 200 $\Omega$m. The inversion of the imaginary part at 40 kHz (Figure 7(c)) reveals higher values from the eastern side extending towards the center, not more than 4 meters deep, as well as on the western side.




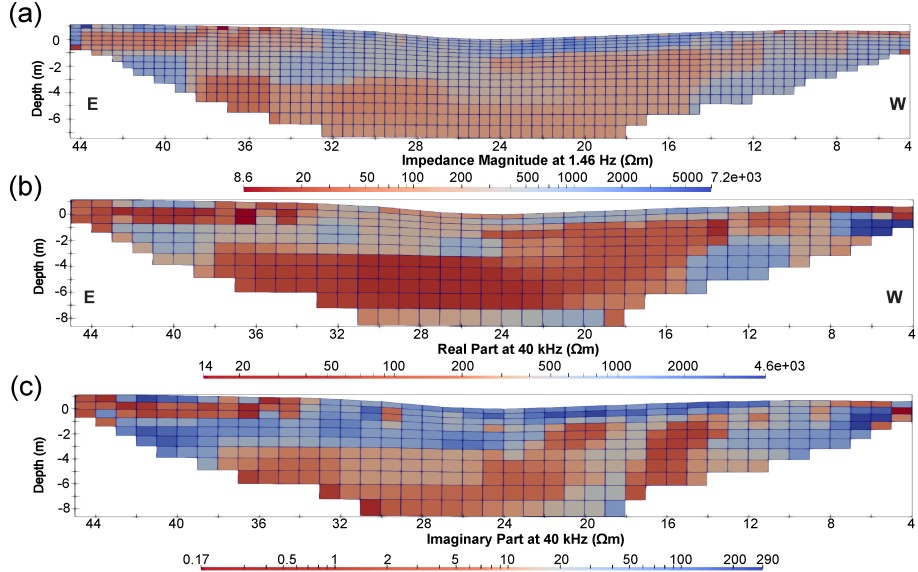

**Figure 7.** The inversion of (a) impedance magnitude at 1.46 Hz, (b) real and (c) imaginary part at 40 kHz above the RTS.

The ERT-like result (Figure 7(a)) provides insight into the ground resistivity distribution, which, in a simple geological environment, can effectively distinguish unfrozen soil from soil with ground ice. However, in some geological settings and temperature conditions, ERT is less reliable for identifying ground ice (Tourei et al., 2024). The inversion at 40 kHz brings more details of the presumed ground ice distribution, with higher values in the imaginary part (Figure 7(c)) potentially indicating the presence of ground ice. Although the inversion result at 40 kHz (Figure 7(c)) offers more detail than ERT-like data (Figure 7(a)), they are not conclusive, either. For instance, high values in the imaginary part do not uniquely identify ground ice, and there is no guarantee that high values at 40 kHz correspond to ground ice.

ART was extracted from the field spectra and assigned to pseudo depths (Figure 8). Data points with relaxation times that fall within the ground ice relaxation time (see: *short profile in headwall Section*) range are shown in blue. The eastern part of the ART profile (Figure 8(a)) at the headwall location shows a shallow frozen layer up to 0.75 meters deep, followed by an unfrozen layer underlain by a mix of frozen and unfrozen layers. The relaxation time histogram (Figure 8(b)) exhibits a multimodal pattern, with the most prominent peaks occurring at 4, 9.6, and 40 $\mu$s. The 4 $\mu$s relaxation time is lower than that of ground ice, and 9.6 $\mu$s lies at the threshold between non-ground ice and ground ice. The average ART value at depths of 2 to 4 meters in the western part (0 to 17 m) is 3.97 $\mu$s, which falls outside the range for ground ice relaxation time. Similarly, the eastern part at the same depth (24 to 44 m) shows an ART value of 1775 $\mu$s, which is also far from ground ice relaxation time.



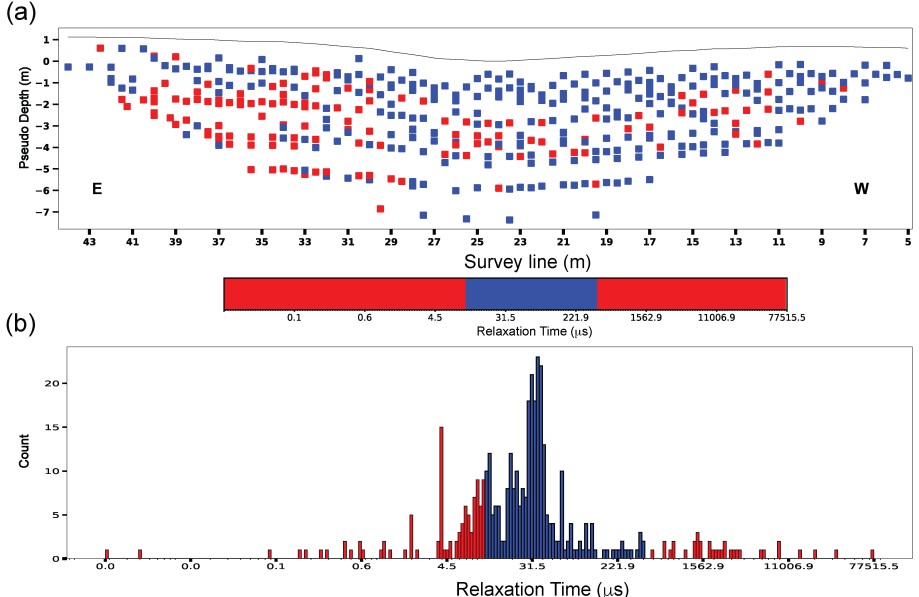

**Figure 8.** (a) Apparent relaxation time (ART) distribution in the long profile above the RTS headwall, and (b) histogram of ART.

## 5.3 Pingo

The inversion results (Figure 9) show a consistent signature of the ice core, extending previous findings that have been corroborated with a borehole (Fereydooni et al., 2024). A high-impedance area is observed to a depth of 0.5 m near the surface, followed by a lower-impedance layer extending from 0.5 to 1 m below, and finally a high impedance area again at depth.

The extracted ART (Figure 10(a)) indicates the approximate distribution of ground ice. Although the individual frequency inversion profile could not identify ice on the east side, the ART profile detected its presence. The mode of the ART is 30 $\mu$s

(Figure 10(b)), a strong sign of the pingo ice core.

Near the pingo apex (at electrode 22), a borehole was drilled in April 2023 to evaluate SIP results (Fereydooni et al., 2024). Ground ice was present from 0.6 m to 8.3 m depth (Figure 11(a)) and ART values align well with this pattern. At a depth of 0.48 m, the ART is 975 $\times 10^3$ $\mu$s, far outside the typical range for ground ice. However, at 0.70 m, the ART shifts to 26 $\mu$s, which is within the range associated with ground ice. From 0.7 m to 3.4 m, ART varies in the range 10–35 $\mu$s. Below 3.4 m,

no ART has been extracted.

The apparent resistivity values (Figure 11(b)) at the borehole location range from 150 to 600 $\Omega$m, which is significantly below the threshold for ground ice identification (>1000 $\Omega$m). These low resistivity values are characteristic of clay-rich sediments, which exhibit higher conductivity due to their surface properties and the presence of absorbed cations in the clay minerals. Despite visible ice content in the clayey silt (5%) and silty clay layers (20-30%), the dominant clay matrix masks the

high resistivity signature typically associated with ground ice. This masking effect makes it challenging to identify ground ice using resistivity methods.



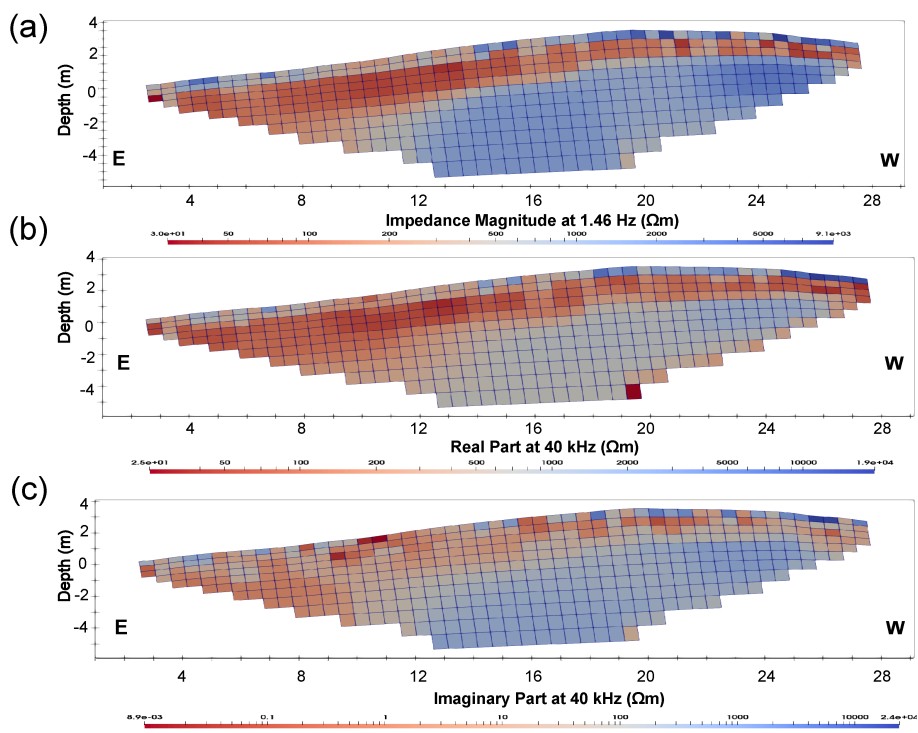

**Figure 9.** The inversion of SIP data over the pingo: (a) impedance magnitude at 1.46 Hz, (b) real and (c) imaginary part at 40 kHz.

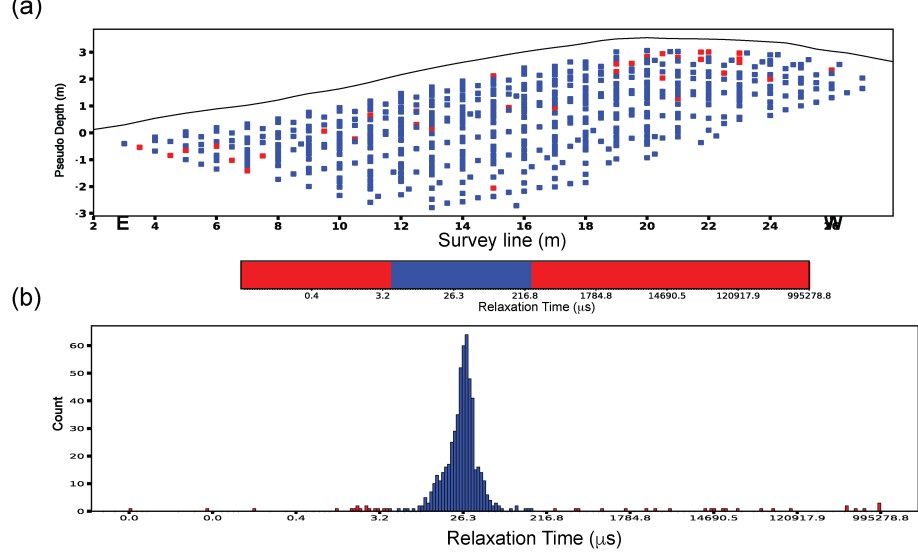

**Figure 10.** (a) ART distribution, and (b) Relaxation time histogram for the pingo site.





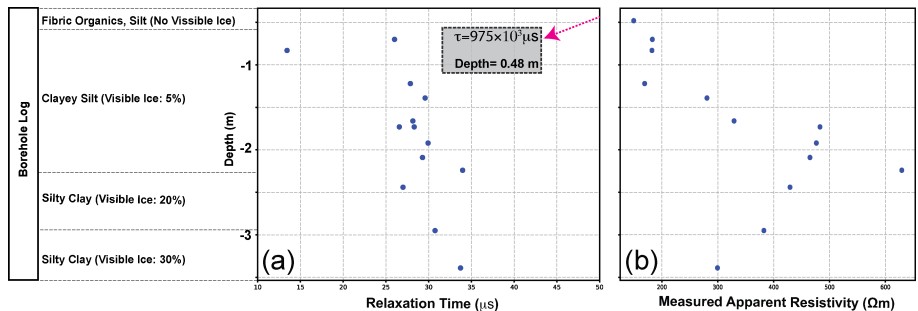

**Figure 11.** (a) Relaxation time and (b) apparent resistivity at the borehole location near the pingo apex at electrode 22.

## 6 Discussion

### 6.1 Retrogressive thaw slump: long profile

Figure 12a shows the ART distribution and the inversion of the imaginary part of SIP data in one plot. Here, we discuss the
available geophysical information as well as that from the exposed slump headwall. Despite their proximity, however, some
ambiguity remains. We may assume that the same frozen material is found beneath the analyzed SIP profile or alternatively
that it has already been thawed there due to its location near the edge of the right-of-way for the Alaska highway. Although
the inversion provides more information than ERT, it is not compatible with the ART distribution. For example, the eastern
part of the inversion profile (yellow box) shows high resistivity (indicating frozen ground) at a depth of around 2 meters
that is not compatible with the exposed headwall stratigraphy (unfrozen silty and sand layers), while the ART would identify
it as unfrozen ground. Additionally, the black box indicates that frozen ground may be misinterpreted as unfrozen ground
(lower resistivity) in the inversion profile (Figure 12(a)). This discrepancy arises because the relaxation time of ground ice is
independent of the magnitude of the imaginary part at a single frequency. For instance, Maierhofer et al. (2022) found that
phase images did not show a clear correlation with borehole data when using frequencies of 0.5 Hz and 75 Hz, indicating that
the imaginary part at these specific frequencies cannot distinctly differentiate ground ice from other materials. This finding
suggests that the imaginary part alone is insufficient for accurately identifying ground ice, as it does not capture the full
spectrum of frequency-dependent behaviors.

Also, the other research has been utilizing other method like Resistivity Frequency Effect (RFE) to consider two individual
frequencies (Grimm et al., 2015; Fereydooni et al., 2024). Considering the real part of apparent resistivity ($\Omega$m) values at the
lowest (1.46 Hz) and highest (40 kHz) frequencies, we can use Eq.(11) for RFE, where a value close to 1 indicates higher ice
content:

$$\text{RFE} = \frac{\rho_{1.46\text{Hz}} - \rho_{40\text{kHz}}}{\rho_{1.46\text{Hz}}} \tag{11}$$





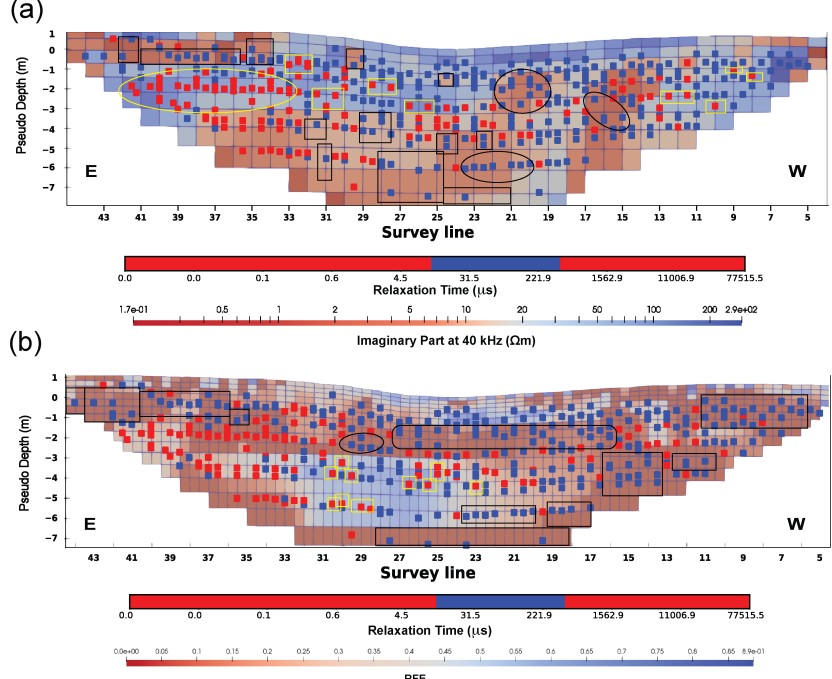

**Figure 12.** ART with (a) the imaginary part at 40 kHz, and (b) RFE for the long profile retrogressive thaw slump study area.

Figure 12b presents the ART distribution and RFE in a single plot. Although the RFE identifies ground ice in the middle of the survey (where ART indicates ground with and without ice, yellow boxes), it indicates presumed ground ice areas as thawed (the black boxes). This difference arises because the RFE relies only on two frequency data points (minimum and maximum), limiting its ability to accurately map ground ice distribution.

In 2019, an ERT profile was measured in the thaw slump study area, using a dipole-dipole array, overlapping with our SIP measurements. The ERT results did not reveal any signs of ground ice, while the SIP survey did detect ground ice at that location and depth. The report suggested that permafrost could exist in areas with resistivity as low as 100 Ωm, possibly due to the presence of fine-grained materials with higher liquid water content (Roy et al., 2021). Since ERT relies solely on resistivity, it may incorrectly interpret bedrock (a high-resistivity material) as ground ice and misidentify ice-rich clay and silt just below 0C (low-resistivity materials) as non-ground ice.

## 6.2 Pingo

Figure 13a illustrates the imaginary part inversion at 40 kHz and the ART distribution. The black box highlights the area where the results of the two methods align. The inversion results indicate most of the eastern part of profile as unfrozen ground, while ART identifies it as frozen ground. Also, Fig. 13b shows the overlap between RFE and ART distribution. The RFE and ART




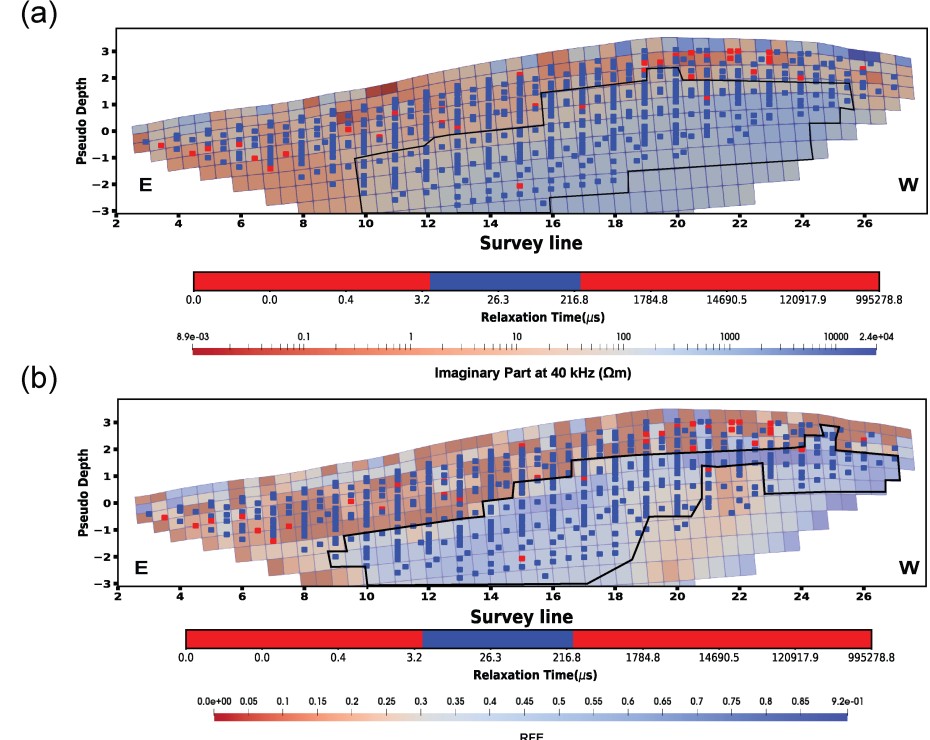

**Figure 13.** ART with (a) Imaginary part at 40 kHz, and (b) RFE for the pingo study area.

distribution do not align well, as lower RFE values (associated with thawed, ice-free zones) contrast with the ART distribution, which corresponds to ground ice relaxation times.

While Fig. 13 compares ART (in pseudo depth) without inversion processing and the imaginary part, and RFE undergoes
inversion (Figure 13), this study deliberately compared paired data of the same nature (non-inverted data) for the pingo site to ensure consistency in analysis. Figure 14 compares imaginary part data at 40 kHz and relaxation time at the pingo site. Imaginary part values exceeding $\bar{x}$(Mean) $+ 1 \times \sigma$(Standard deviation) $(10^3 \ \Omega m)$ are considered anomalies. Ground ice detection is based on imaginary part values above this threshold and relaxation times between 10–400 $\mu s$.

Figure 14a shows the histogram of the imaginary part at 40 kHz, with most data concentrated below 100 $\Omega m$. Figure 14b
quantifies the effectiveness of different detection criteria. The results show that 83.6% of data are recognized as ground ice using relaxation time measurements alone (green sector), while 7.9% were identified as ground ice using both relaxation time and imaginary part criteria (blue sector), 8.1% of data were not detected as ground ice (gray sector), and only 0.3% of detections relied solely on the imaginary part criterion (red sector). Figure 14c displays a scatter plot correlating imaginary part values (x-axis) with relaxation times (y-axis) on logarithmic scales. Dashed lines mark thresholds: relaxation times between 10–400
$\mu s$ and imaginary part >103 $\Omega m$.





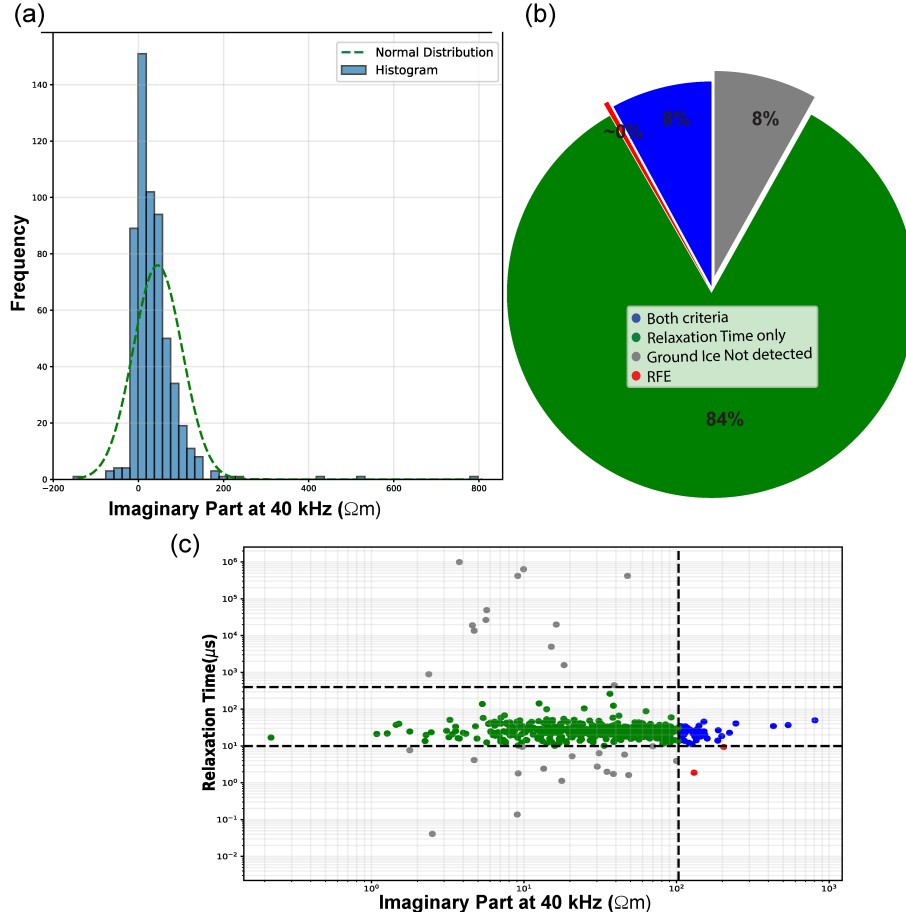

**Figure 14.** Comparison of relaxation time and imaginary part at 40 kHz in the pingo site: (a) Histogram of imaginary part values, (b) Pie chart of ground ice detection criteria, and (c) Scatter plot of imaginary part vs. relaxation time.

This analysis demonstrates, based on the defined criteria, that relaxation time measurement is the dominant and more reliable indicator for ground ice detection in this pingo site, with the imaginary part criterion providing complementary information in a smaller subset of cases.

Figure 15 compares RFE data and relaxation time measurements at the pingo site. RFE values exceeding $\bar{x} + 1 \times \sigma$ (approx-
imately 0.2) are considered anomalies. Ground ice detection is based on RFE values above this threshold and relaxation times between 10–400 $\mu$s.

Figure 15a shows RFE histogram, with most data concentrated between 0.05 and 0.25. Figure 15b provides a pie chart quantifying the effectiveness of different detection criteria. The results show that 65.5% of data are recognized as ground ice using relaxation time measurements alone (green sector), while 29.5% were identified as ground ice using both relaxation time and RFE criteria (blue sector), 3.9% of data were not detected as ground ice (gray sector), and only 1.1% of detections relied



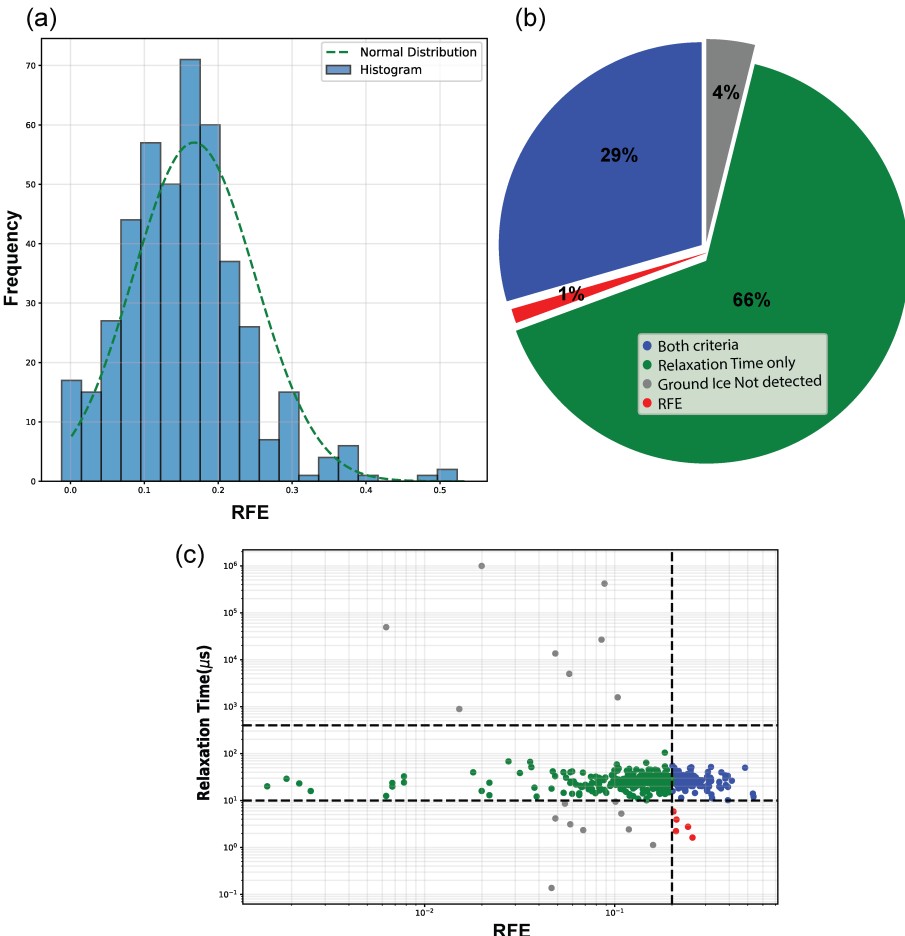

**Figure 15.** Comparison of relaxation time and RFE in the pingo site: (a) Histogram of RFE values, (b) Pie chart of ground ice detection criteria, and (c) Scatter plot of RFE vs. relaxation time.

solely on the RFE criterion (red sector). Figure 15c displays a scatter plot correlating RFE values (x-axis) with relaxation times (y-axis) on logarithmic scales. Dashed lines mark thresholds: relaxation times between 10–400 $\mu$s and RFE >0.2.

This analysis demonstrates, based on the defined criteria, that relaxation time measurement remains the dominant indicator for ground ice detection in the pingo site, with the RFE criterion providing substantial complementary information in nearly 345 30% of cases.

Compared to other methods, SIP can detect ground ice with less ambiguity. Additionally, because relaxation time is related to ground ice temperature and ice content, further research could potentially link SIP data to these critical ground ice properties. Despite these advantages, SIP is a relatively new method, and its conceptual understanding, as well as the availability of devices for both field and laboratory measurements, is still limited. Field data acquisition with SIP takes longer than conventional



methods like ERT, as each layout must be configured manually. Furthermore, data processing and inversion techniques for SIP are still underdeveloped and require additional research.

## 7 Conclusions

Due to the limitations of electrical resistivity methods in identifying ground ice (where underground ice in clay and highly conductive materials, with low resistivity, can be misidentified as unfrozen ground, and bedrock, with high resistivity, can be
misidentified as ground ice), this study explored the Spectral Induced Polarization (SIP) method. SIP measures both the real and imaginary parts of impedance across a range of frequencies. This study demonstrated that relying on a single frequency (the imaginary part) or on two frequencies (RFE) can lead to unreliable conclusions. Because RFE only considers the difference between minimum and maximum frequencies in the presence of conductive materials, it fails to identify ground ice, with values tending toward zero. Similarly, high values in the imaginary part do not uniquely indicate ground ice, and there is no guarantee
that these high values correspond to ground ice. Instead, relaxation time, which is derived using all measured frequencies, was utilized as a distinctive indicator of ground ice. The relaxation time range indicative of ground ice was determined from field measurements and previous research, then applied to extracted ART in two case studies, one at an RTS and one on a pingo. The SIP results were interpreted and contextualized with a natural exposure of permafrost in an RTS headwall near an SIP profile, and logs from a borehole drilled in the pingo SIP profile. The interpretation based on relaxation time is a method of
analysis that will be even more relevant once SIP can be properly inverted. While this study did not focus on the relationship between relaxation time, ice content, and temperature, these factors are important and warrant further investigation. Despite its advantages, SIP is limited by its relative novelty, a lack of accessible equipment, longer field setup times compared to ERT, and underdeveloped data processing and inversion techniques that require further research.

*Author contributions.* Conceptualization of the work was carried out by Hosein Fereydooni and Stephan Gruber. Fieldwork, investigations,
formal analysis, and visualizations were carried out by Hosein Fereydooni, Stephan Gruber, and Derek Cronmiller. David Stillman provided expert review on the methodology, results, and processing of the data. Paper writing and editing were led by Hosein Fereydooni and completed by all authors.

*Competing interests.* The authors declare that they have no known competing financial interests or personal relationships that could have appeared to influence the work reported in this paper.

*Acknowledgements.* This work would not have been possible without the help of Moya Painter, and Patrick Sack. Funding for this field campaign came from the Yukon Geological Survey (contribution 065) and the Natural Science and Engineering Research Council of Canada via NSERC PermafrostNet (NETGP 523228-18) and Discovery Grant RGPIN-2020-04783.




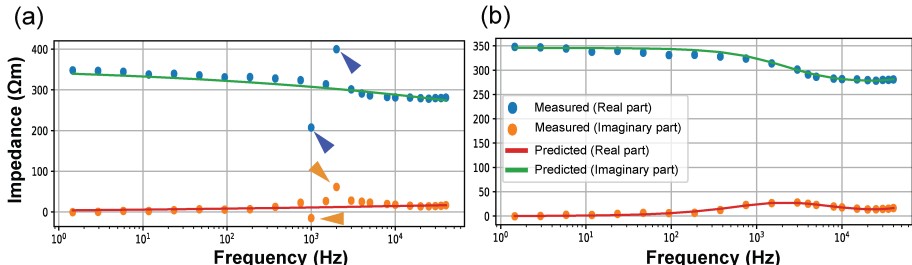

**Figure A1.** (a) SIP data with spike noise at 1 and 2 kHz, along with the predicted model (b) SIP data and predicted model after noise elimination.

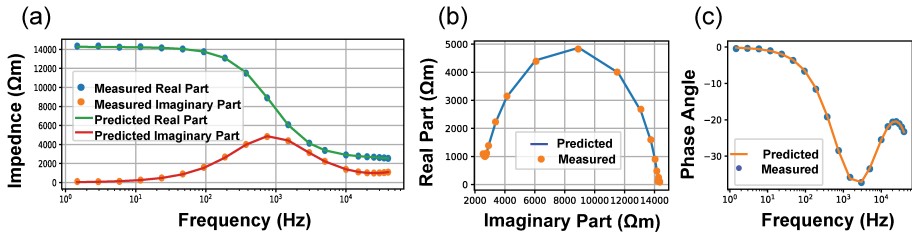

**Figure A2.** Measured and predicted SIP data, showing (a) real and imaginary components, (b) Nyquist plot, and (c) phase angle data for the short horizontal survey.

## Appendix A: Fitting Example

Figure A1 shows the ability of the code to exclude two noisy frequencies at 1 and 2 kHz (Figure A1(a)), and then, to fit a

multi-Cole-Cole model (Figure A1(b)).

Figure A2 displays measured and predicted SIP data, with a relaxation frequency of 188 $\mu$s and an RMSE of 0.42%.

Table A2 shows the multi Cole-Cole parameters used to plot Figure 2.



**Table A1.** Relaxation time of frozen soil samples at various temperatures; which Sample 1: 84.3% sand, 12.5% silt, and 3.2% clay; Sample 2: 8.3% sand, 78.4% silt, and 13.3% clay; and Sample 3: 17.9% sand, 36.5% silt, and 45.6% clay. (Bittelli et al., 2004)

| Temperature ($^{\circ}C$) | $\tau(Sample_1)$ $\mu$s | $\tau(Sample_2)$ $\mu$s | $\tau(Sample_3)$ ($\mu$s) |
|---|---|---|---|
| -30 | 82.46 | 52.79 | 47.23 |
| -25 | 79.10 | 49.44 | 42.51 |
| -20 | 64.36 | 44.66 | 37.60 |
| -15 | 50.99 | 36.78 | 33.22 |
| -10 | 36.34 | 32.11 | 29.96 |
| -5 | 32.06 | 28.61 | 25.49 |
| -3 | 29.87 | 26.58 | 24.04 |
| -1 | 26.92 | 21.35 | 20.06 |

**Table A2.** Parameters used for multi Cole-Cole models in Fig. 2 (Behari, 2005; Bittelli et al., 2004).

| Parameter | Value(s) |
|---|---|
| $\rho_0$ | 100 |
| $m_1$ (double Cole-Cole) | 0.283 |
| $m_2$ (double Cole-Cole) | 0.370 |
| $\tau_1$ ($\mu$s) for unfrozen soil | 0.614, 0.619, 0.922, 0.664, 0.618, 0.614, 0.622, 0.371 |
| $\tau_2$ ($\mu$s) for frozen soil | 82.464, 79.103, 64.357, 50.995, 36.345, 32.062, 29.871, 26.925 |
| $c_1$ | 1.109 |
| $c_2$ | 0.858 |
| Frequency range | 1 Hz to 1 MHz (50 logarithmically spaced points) |

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
