# Peer review of "Detecting ground ice in warm permafrost with the dielectric relaxation time from SIP observations"

_EGUsphere, 2025_

## Referee Comment (RC1)

Review of the manuscript entitled: "Detecting ground ice in warm permafrost with the dielectric relaxation time from SIP observations"

I thoroughly enjoyed reading this article. It is written in a very clear and logical manner, presenting a good field application for SIP data and how it can be used to distinguish areas of the subsurface which contain ground ice.

Overall, I believe the article is in good shape but requires a bit more technical detail, particularly on error reporting, and is missing a couple of elements for a complete discussion. I have some suggestions and comments for the authors which I will enumerate below. The comments I found to require more attention and are critical for the narrative and consistency of this work are written in red.

Line 52: Real part is resistance, which is different from resistivity. It is important to have the right terminology, especially in this (theory) section. You may wish to explain the relationship between resistance and resistivity after.

Table 1: Is this produced with Eq. 7? If so, it is not very clear to the reader.

Line 189: Are the ground electrodes Ag/AgCl? This type of electrodes is used in SIP surveys to avoid polarization. Please specify.

Line 213: It looks like you have accurate GPS coordinates of the electrodes used. Are these being used in your inversions?

Figure 6: X and Y axis font is too small.

Methods and Results: There is not a great deal of detail about post-acquisition data processing. You also don't report any error metrics corresponding to inversion or ART results. I believe this needs to be addressed in order to contextualise the accuracy of the model results presented.

Consequently, the error corresponding to your results could have an impact on which areas you can confidently categorize as "containing ground ice".

Figure 11: Can you comment on the fact that relaxation time does not seem to change when ice content increases. It seems to be effective in picking up presence of ice but not disgusting between different levels of ice content.

Line 298: Remove "the"?

Line 300: I would probably introduce RFE earlier, alongside the other equations in your methodology, not in the discussion.

Figure 12: Please mention what yellow and black outlines mean in figure caption.

Figure 13: As for figure 12, please explain the black outlines in the figure caption.

Line 312: Degree symbol.

Line 330: How was the threshold for the imaginary part determined?

Section 6.2: I don't fully understand the logic behind figures 14 and 15. ART categorizes some datapoints as in the "contains ground ice region", Imaginary part does the same, and there is some overlap between the two. However, how do you know which one is correct? Is this based on the borehole data? Are you assuming everything below 0.6m depth (borehole log) has an above zero percentage of ground ice?

Line 367: I suggest you expand this is a short paragraph in your discussion. The section would be about the limitations of the SIP method, pathways for improvement and future research. I believe this would round up your narrative and provide prospects for future studies.

---

## Author Comment (AC1)

[revised manuscript text omitted]

The measured spectra are evaluated for spike noise, identified using the $3\sigma$ rule (values exceeding three standard deviations from either the mean impedance magnitude or the mean phase angle), which may occur at one or more frequencies. In such cases, the data points identified as outliers are excluded specifically at the corresponding frequencies, a step that reduces sensitivity to the initial guess and allows for a broader range of possible values for each unknown parameter, rather than restricting the search to a narrow range (Appendix Fig. A1). If a positive phase angle is detected, it will be investigated whether these values result from unfrozen material or are indicative of spike noise, and it will follow the same procedure outlined for handling spike noise.

**3.2 Extracting relaxation times**

Extracting relaxation times is one of the avenues for processing SIP data to allow interpretation. It is challenging, because measured spectra have noise that can complicate the extraction of parameters, and more than one relaxation feature may affect a frequency band of interest. In this study, we fit a two Cole-Cole model (Eq. (12)) to SIP spectra to extract seven unknown parameters for the two most prominent relaxation frequencies (Pelton et al., 1978).

$$\rho^*(\omega) = \rho_0 \left\{ 1 - \sum_{r=1}^{2} m_r \left[ 1 - \frac{1}{1 + (i\omega\tau_r)^{c_r}} \right] \right\} \tag{12}$$

Where $\omega$ represents the angular frequency, $\rho^*(\omega)$ is the apparent complex resistivity, $\rho_0$ is DC resistivity at low frequency, $m_1$ and $m_2$ are the chargeabilities, $\tau_1$ and $\tau_2$ are the relaxation time, $c_1$ and $c_2$ are Cole-Cole exponents for relaxation 1,and 2. Utilizing the two Cole-Cole model, which extracts two relaxation times here, allows for identifying relaxation times of mixed materials, such as soils. For instance, the combination (Fig. 2e, and f) of frozen soil samples (Fig. 2a, and b) and unfrozen soil samples (Fig. 2c, and d), (see Appendix Table A2 for the parameters values) that cannot be explained by one single relaxation time (Eq.(6)).

We utilized a general least square method to minimize the difference between measured SIP spectra, and data predicted by the two Cole-Cole model (Eq.(12)) to find seven unknown parameters ($\rho_0$, $m_1$, $m_2$, $\tau_1$, $\tau_2$, $c_1$, and $c_2$). For this process, we developed a Python script. It first converts the field-measured impedance to apparent complex resistivity by using geometric factor, then assigns a weight to the real and imaginary part of data to ensure those components are fairly balanced and manages noisy data using device-reported error percentages (The device provides separate uncertainties for the real and imaginary parts. We use these to assign inverse weights so that lower error, stronger signal points have more influence).

The two extracted relaxation times ($\tau_1$, $\tau_2$) will then be compared to the ground ice relaxation time range. If either of them falls within this range, it will be identified as ground ice SIP spectrum.

[Figure]

**Figure 2.** Synthetic data (Eq. 12, Appendix Table A2) showing SIP spectra for frozen soil (a, b); unfrozen soil (c, d); and mixed soil (e, f). Panels (a), (c), and (e) show the real (solid line) and imaginary (dashed line) parts; panels (b), (d), and (f) show the corresponding phase angle in degree.

In this paper, the accuracy of the results is presented using the Root Mean Square Error (RMSE $< 6\%$) percentage (The RMSE is calculated on the magnitude, which combines real and imaginary parts into a single measure and avoids scale/unit differences between them):

$$\text{RMSE Percentage} = \frac{\sqrt{\frac{1}{n}\sum_{k=1}^{n}(|Z|_{\text{m},k} - |Z|_{\text{f},k})^2}}{\overline{|Z|}} \times 100 \tag{13}$$

where $|Z|_{\text{m},k}$ and $|Z|_{\text{f},k}$ are the measured and predicted impedance magnitudes for the $k$-th frequency, respectively, and $\overline{|Z|}$ is the mean of the measured impedance magnitudes.

While the RMSE percentage can provide useful insights for determining the reliability of the results of the spectrum fit from the code, the results still need to be evaluated by an expert because the optimization process might get trapped in a

[Figure]

**Figure 3.** Comparison of synthetic Voigt circuit data and predicted results to test the code. (a) Real and imaginary components, and (b) Phase angle. Dots represent the synthetic Voigt model values and lines show predicted values.

local minimum, producing a low RMSE but not necessarily the best possible fit. An expert can recognize when this occurs by examining the overall shape of the fitted curves and their physical plausibility.

The apparent relaxation time (ART) is then assigned a pseudo depth calculated with the ResIpy software package (Blanchy et al., 2020) to provide a 2D image of the distribution of ART versus pseudo-depth. The concept of pseudo-depth in electrical resistivity surveys represents an approximation of the depth range from which the majority of the measured signal originates. Typically, about 70-80% of the signal comes from this range, although this can vary depending on the array geometry and subsurface conditions (Edwards, 1977). The vertical sensitivity of these measurements follows a bell-curve distribution, with the peak contribution occurring at the pseudo-depth (Roy, 1974).

**3.3 Code testing**

We evaluate the method and code using synthetic data from a Voigt model, which represents multiple relaxations consisting of parallel resistor-capacitor (RC) elements, each representing a distinct relaxation mechanism. We use $R_1 = 1\,\mathrm{k\Omega}$, $R_2 = 10\,\mathrm{k\Omega}$, $C_1 = 50\,\mathrm{mF}$, $C_2 = 0.5\,\mathrm{mF}$. To simulate imperfect observations, we add five percent uniform noise.

The relaxation times predicted from the proposed code are $\tau_1$= 0.25s and $\tau_2$= 0.005s with RMSE=$3.5 \times 10^{-5}$%, which match the exact relaxation times of the Voigt model $\tau_1 = R_1 \times C_1$= 0.25s, and $\tau_2 = R_2 \times C_2$= 0.005s as shown in Fig. 3. This demonstrates the ability of the proposed code to extract the relaxation times.

**3.4 Instrumentation**

[revised manuscript text omitted]
 (The black line outlines data points that are identified as containing ground ice by ART but as unfrozen based on the imaginary part at 40 kHz (a) and RFE (b) profiles. The yellow line indicates data points identified as unfrozen by ART but as containing ground ice by the imaginary part at 40 kHz (a) and RFE (b) profiles).

it may incorrectly interpret bedrock (a high-resistivity material) as ground ice and misidentify ice-rich clay and silt just below 0°C (low-resistivity materials) as non-ground ice.

**6.2 Pingo**

Figure 13a illustrates the imaginary part inversion at 40 kHz and the ART distribution. The black box highlights the area where
325     the results of the two methods align. The inversion results indicate most of the eastern part of profile as unfrozen ground, while ART identifies it as frozen ground. Also, Fig. 13b shows the overlap between RFE and ART distribution. The RFE and ART distribution do not align well, as lower RFE values (associated with thawed, ice-free zones) contrast with the ART distribution, which corresponds to ground ice relaxation times.

      While Fig. 13 compares ART (in pseudo depth) without inversion processing and the imaginary part, and RFE undergoes
330     inversion (Figure 13), this study deliberately compared paired data of the same nature (non-inverted data) for the pingo site to ensure consistency in analysis. Figure 14 compares imaginary part data at 40 kHz and relaxation time at the pingo site.

[Figure]

**Figure 13.** ART with (a) Imaginary part at 40 kHz, and (b) RFE for the pingo study area (the black outline highlight the area where the ART and the imaginary part at 40 kHz (a) and RFE (b) profiles are in agreement).

Imaginary part values exceeding $\bar{x} + 1 \times \sigma$ (where $\bar{x}$ is mean, and $\sigma$ is 
[revised manuscript text omitted]

Evans, S.: Dielectric Properties of Ice and Snow–a Review, Journal of Glaciology, 5, 773–792, https://doi.org/10.3189/s0022143000018840, 1965.

Farzamian, M., Herring, T., Vieira, G., de Pablo, M. A., Yaghoobi Tabar, B., and Hauck, C.: Employing automated electrical resistivity to-
415 mography for detecting short- and long-term changes in permafrost and active-layer dynamics in the maritime Antarctic, The Cryosphere, 18, 4197–4213, https://doi.org/10.5194/tc-18-4197-2024, 2024.

Fereydooni, H., Gruber, S., Cronmiller, D., and Stillman, D.: Utilizing spectral induced polarization to identify the ice core of a pingo: a case study in Haines Junction, Yukon, Canada, International Conference on Permafrost, https://doi.org/10.52381/ICOP2024.211.1, 2024.

Flemal, R. C.: Pingos and pingo scars: Their characteristics, distribution, and utility in reconstructing former permafrost environments,
420 Quaternary Research, 6, 37–53, https://doi.org/https://doi.org/10.1016/0033-5894(76)90039-9, 1976.

Grimm, R. E. and Stillman, D. E.: Field test of detection and characterisation of subsurface ice using broadband spectral-induced polarisation, Permafrost and Periglacial Processes, 26, 28–38, https://doi.org/10.1002/ppp.1833, 2015.

Grimm, R. E., Stillman, D. E., and MacGregor, J. A.: Dielectric signatures and evolution of glacier ice, Journal of Glaciology, 61, 1159–1170, https://doi.org/10.3189/2015JoG15J113, 2015.

425 Hauck, C., Böttcher, M., and Maurer, H.: A new model for estimating subsurface ice content based on combined electrical and seismic data sets, The Cryosphere, 5, 453–468, 2011.

Li, X., Jin, X., Wang, X., Jin, H., Tang, L., Li, X., He, R., Li, Y., Huang, C., and Zhang, S.: Investigation of permafrost engineering geological environment with electrical resistivity tomography: A case study along the China-Russia crude oil pipelines, Engineering Geology, 291, https://doi.org/10.1016/j.enggeo.2021.106237, 2021.

430 Limbrock, J. K. and Kemna, A.: Relationship between Cole-Cole model parameters in permittivity and conductivity formulation, Geophysical Journal International, 239, 964–970, https://doi.org/10.1093/gji/ggae300, 2024.

Maierhofer, T., Hauck, C., Hilbich, C., Kemna, A., and Flores-Orozco, A.: Spectral induced polarization imaging to investigate an ice-rich mountain permafrost site in Switzerland, The Cryosphere, 16, 1903–1925, https://doi.org/10.5194/tc-16-1903-2022, 2022.

Maierhofer, T., Orozco, A. F., Roser, N., Limbrock, J. K., Hilbich, C., Moser, C., Kemna, A., Drigo, E., Cella, U. M. D., and Hauck, C.:
435 Spectral induced polarization imaging to monitor seasonal and annual dynamics of frozen ground at a mountain permafrost site in the Italian Alps, Cryosphere, 18, 3383–3414, https://doi.org/10.5194/tc-18-3383-2024, 2024.

Mudler, J., Hördt, A., Kreith, D., Sugand, M., Bazhin, K., Lebedeva, L., and Radić, T.: Broadband spectral induced polarization for the detection of permafrost and an approach to ice content estimation - a case study from Yakutia, Russia, The Cryosphere, 16, 4727–4744, https://doi.org/10.5194/tc-16-4727-2022, 2022.

440 Musil, M., Maurer, H., Green, A. G., Horstmeyer, H., Nitsche, F. O., Mühll, D. V., and Springman, S.: Shallow seismic surveying of an Alpine rock glacier, Geophysics, 67, 1701–1710, https://doi.org/10.1190/1.1527071, 2002.

Oldenborger, G. A.: Subzero temperature dependence of electrical conductivity for permafrost geophysics, Cold Regions Science and Technology, 182, https://doi.org/10.1016/j.coldregions.2020.103214, 2021.

Pelton, W. H., Ward, S. H., Hallof, P. G., Sill, W. R., and Nelson, P. H.: Mineral discrimination and removal of inductive coupling with multifrequency IP, Geophysics, 43, 588–609, https://doi.org/10.1190/1.1440839, 1978.

Petrenko, V. F. and Whitworth, R. W.: Physics of ice, OUP Oxford, 1999.

Revil, A., Razdan, M., Julien, S., Coperey, A., Abdulsamad, F., Ghorbani, A., Gasquet, D., Sharma, R., and Rossi, M.: Induced polarization response of porous media with metallic particles - Part 9: Influence of permafrost, Geophysics, 84, E337–E355, https://doi.org/10.1190/geo2019-0013.1, 2019.

Roy, A.: Depth of investigation in Wenner, three-electrode and dipole-dipole DC resistivity methods, Geophysics, 39, 200–206, https://doi.org/10.1190/1.1440421, 1974.

Roy, L.-P., Calmels, F., Laurent, C., Vogt, N., Lipovsky, P. S., Humphries, J., and Survey, Y. G.: Greater Whitehorse area permafrost characterization. Yukon Geological Survey, Miscellaneous Report MR-22, 185 p., including appendices, https://data.geology.gov.yk.ca/Reference/95911#InfoTab, 2021.

Stillman, D. E., Grimm, R. E., and Dec, S. F.: Low-frequency electrical properties of ice-silicate mixtures, Journal of Physical Chemistry B, 114, 6065–6073, https://doi.org/10.1021/jp9070778, 2010.

Stillman, D. E., MacGregor, J. A., and Grimm, R. E.: The role of acids in electrical conduction through ice, Journal of Geophysical Research: Earth Surface, 118, 1–16, https://doi.org/10.1029/2012JF002603, 2013a.

Stillman, D. E., MacGregor, J. A., and Grimm, R. E.: Electrical response of ammonium-rich water ice, Annals of Glaciology, 54, 21–26, https://doi.org/10.3189/2013AoG64A204, 2013b.

Tourei, A., Ji, X., dos Santos, G. R., Czarny, R., Rybakov, S., Wang, Z., Hallissey, M., Martin, E. R., Xiao, M., Zhu, T., Nicolsky, D., and Jensen, A.: Mapping permafrost variability and degradation using seismic surface waves, electrical resistivity, and temperature Sensing: A case study in arctic Alaska, Journal of Geophysical Research: Earth Surface, 129, https://doi.org/10.1029/2023JF007352, 2024.

Wagner, F. M., Mollaret, C., Günther, T., Kemna, A., and Hauck, C.: Quantitative imaging of water, ice and air in permafrost systems through petrophysical joint inversion of seismic refraction and electrical resistivity data, Geophysical Journal International, 219, 1866–1875, 2019.

You, Y., Yu, Q., Pan, X., Wang, X., and Guo, L.: Application of electrical resistivity tomography in investigating depth of permafrost base and permafrost structure in Tibetan Plateau, Cold Regions Science and Technology, 87, 19–26, https://doi.org/10.1016/j.coldregions.2012.11.004, 2013.

Zorin, N. and Ageev, D.: Electrical properties of two-component mixtures and their application to high-frequency IP exploration of permafrost, Near Surface Geophysics, 15, 603–613, https://doi.org/10.3997/1873-0604.2017043, 2017.